# Accelerating Federated Learning Through Attention on Local Model Updates

**Parsa Assadi, Byung Hoon Ahn, Hadi Esmaeilzadeh**
University of California, San Diego
{passadi, bhahn221, hadi}@ucsd.edu

## Abstract

Federated learning is used widely for privacy-preserving training. It performs well if the client datasets are both balanced and IID. However, in real-world settings, client datasets are non-IID and imbalanced. They may also experience significant distribution shifts. These non-idealities can hinder the performance of federated learning. To address this challenge, the paper devises an attention-based mechanism that learns to attend to different clients in the context of a reference dataset. The reference dataset is a test dataset in the central server which is used to monitor the performance metric of the model under training. The innovation is that the attention mechanism captures the similarities and patterns of a batch of clients' model drifts (received by the central server in each communication round) in a low dimensional latent space, similar to the way it captures the mutual relation of a batch of words (a sentence). To learn this attention layer, we devise an autoencoder whose input/outputs are the model drifts and its bottleneck is the attention mechanism. The attention weights in the bottleneck are learned by utilizing the attention-based autoencoder as a network to reconstruct the model drift on reference dataset, from the batch of received model drifts from clients in each communication round. The learned attention weights effectively capture clusters and similarities amongst the clients' datasets. The empirical studies with MNIST, FashionMNIST, and CIFAR10 under a non-IID federated learning setup show that our attention-based autoencoder can identify the cluster of similar clients. Then the central server can use the clustering results to devise a better policy for choosing participants clients in each communication round, thereby reducing the communication rounds by up to 75% on MNIST and FashionMNIST, and 45% on CIFAR10 compared to FedAvg.

## 1 Introduction

Federated learning is used widely for privacy-preserving training [1]. While it has shown to yield promising results under IID setting, it suffers from slow convergence rate when private datasets are non-IID or class imbalanced [2]. Imbalanced private datasets in the clients may potentially bias the global model as the training progresses. To make the training more robust to unbalanced and non-IID settings, federated learning adopts a policy to select clients in each communication round. For instance, if a client with a certain dominant class is chosen too often due to a bad policy, the global model can become biased towards that class. Therefore, the policy for choosing the client plays an important role in the convergence rate of the global model, and bad policy may prolong the training. While the central server having more information about the distribution of the private datasets can improve the policy, this information is limited to the clients' model drifts as anything more than this would contradict the purpose of federated learning which is to keep the clients' data private. This work sets out to adopt an *attention mechanism* to extract knowledge about private distributions from the model drifts. Inspired by its prevalent adoption in language understanding in which the attention learns the semantic correlation of words in a sentence [3], we utilize the attention mechanism to learn the similarity of private dataset distributions in the context of a reference dataset. The reference dataset is a test dataset used by the central server to monitor the test accuracy of the model throughout the training. We train the attention based autoencoder, so it learns to first reduce dimensionality of model drifts, then uses a multi-headed self attention layer on model drifts in this latent low dimensional space to assign a number to all mutual model drifts, finally, it uses the output of this self attention layer to revert back model drifts to the original high dimensional space and tries to reconstruct the reference model drift from them. After we trained the attention based autoencoder, we can utilize it by feeding a batch of model drifts received by the central server in each communication round

to it, and using the generated weight matrix in the attention layer to cluster them with respect to the private distribution each of them represents. This clustering knowledge is used to adopt a better policy for choosing participants in communication rounds; hence accelerating federated learning. To this end, the paper makes the following contributions: (1) Proposing a framework using an *attention based autoencoder* which only uses model drifts received by the central server to cluster the clients with respect to their private distribution. (2) Devising a *policy* to leverage the clustering knowledge to choose the participants in each communications round to speed up the convergence, reducing the communication rounds and accelerating federated learning. The empirical studies with MNIST, FashionMNIST, and CIFAR10 under a non-IID federated learning setup show that our attention-based autoencoder can identify the cluster of similar clients, and use this knowledge to reduce communication rounds by up to 75% on MNIST and FashionMNIST, and 45% on CIFAR10 compared to FedAvg.

## 2 Related works

Federated learning is used widely for privacy-preserving training [4, 5, 6, 7, 8, 9]; local model updates are the only source of information the central server receives from each client. In fact, information beyond the model drifts may create privacy concerns [10, 11, 12, 13] In real-world settings, non-idealities (e.g., non-IID settings) are inevitable, and they may limit the performance of federated learning. Works including [14] try to mitigate the effect of such non-idealities. This paper explores the use of attention mechanism [15] to (1) cluster the clients to devise better policy to select participants leading to better convergence, thereby (2) less communication. Below, we discuss the most related works.

**Clustering clients for federated learning.** Clustering is one of the solutions used by prior works to deal with statistical non-idealities of clients. Prior works have utilized hierarchical clustering [16], K-means [17], cosine similarity [18] to cluster the clients based on their distribution. [19] utilizes time zone information to better cluster the client. [20] uses spectral analysis to cluster them in a low dimensional latent space, and [21] theoretically studied the statistical heterogeneity issue of clients, and how clustering them can help in reducing the communication cost. In contrast, this work uniquely leverages attention mechanism by considering the batch of model drifts as a batch of words, and trains the whole architecture (dimensionality reduction, clustering) end-to-end for more robust clustering of clients.

**Communication reduction in federated learning.** As communication is the major bottleneck in federated learning, a handful of works [22, 20, 23, 24, 19] have focused on communication reduction. These methods span from reducing the frequency of communication to compressing local updates. However, the focus of this paper, leveraging statistical properties of private distributions to cluster clients, remains orthogonal to these inspiring efforts.

## 3 Challenges in Federated Learning

This section outlines three challenges in federated learning.

**Blindness to clients' datasets in federated learning.** As per the definition of federated learning, each dataset of the clients must be kept *private* to the client. Therefore we rely on information extracted from client updates to create a policy for choosing participants in each communication round. This policy determines the performance of the federated learning and even the convergence rate. To provide the policy with more information about the clients, some works [16, 17, 18] use the information in client updates to cluster clients with respect to

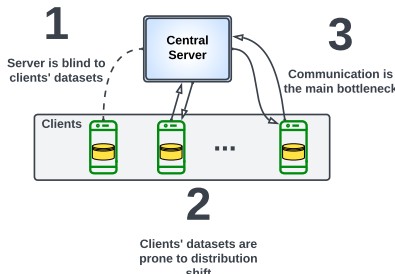

Figure 1: Challenges in federated learning.

their distribution (*knowledge extraction*). From a practical point of view, however, learning from high-dimensional model updates is challenging both from a statistical point view [25], and in terms of compute power.

**Dynamic real-world setting in federated learning.** To make this worse, federated learning may face many non-idealities. Federated learning with IID and balanced datasets is well studied [9]. However, in a real-world setting, the datasets are not only non-IID and imbalanced, but are prone to significant distribution shifts [26]. This leads to slow convergence rate [26, 27]. Even if we manage to adapt the client settings to the non-IID setting at a certain point, this setting changes constantly and frequently rendering this futile. Therefore, it is critical that the policy can incorporate a mechanism that is robust to the above-mentioned dynamic settings while estimating the client setting.

**Communication cost in federated learning.** Even in modern communication systems, communication cost remains the major bottleneck in federated learning [22, 23, 24, 28]. This makes it crucial to finish training the model as soon as possible while achieving the desired performance metric to minimize the *costly* model updates. The policy used for choosing participants directly impacts convergence rate, thus communication cost [27, 28]. Figure 1 summarizes the aforementioned issues in federated learning. Overall, an optimal policy

can be the panacea to all the above challenges, and the optimality of the policy is highly correlated to the effectiveness of the knowledge extraction mechanism that is integrated into the policy. To this end, this work focuses on developing a more effective and robust knowledge extraction mechanism.

## 4  Accelerating Federated Learning Through Attention on Local Model Updates

This section first outlines the insights that drive the design of our knowledge extraction mechanism. Then, we dive into the details of the architecture, training, and its usage in the overall federated learning scenario.

### 4.1  Insights

As discussed in the previous section, the policy for choosing the participants in each communication round determines the performance of the federated learning. However, despite its importance in improving the policy, the information about the clients available to the server is limited to the model updates as per the definition of federated learning. To this end, the problem reduces down to *how we can build a better knowledge extraction framework that can better understand the client settings*. The role of the knowledge extraction framework is two folds: (1) *dimensionality reduction of the model drifts* and (2) *clustering the clients with similar distribution*. Then, to intelligently select the participants for better convergence of federated learning instance, we cluster the clients with regard to their dataset distribution. Prior works utilize conventional methods such as Principal Component Analysis (PCA) and spectral analysis for dimensionality reduction, and K-means and hierarchical clustering to cluster the clients. However, as each of these *conventional methods are not robust* to the aforementioned non-idealities such as distribution shifts [29, 30, 31, 32], the policy that utilizes these methods comes with limited robustness. Furthermore, prior works take a decoupled design where the dimensionality reduction and clustering are two *separate components*. Therefore, similar to how conventional component-based systems only provided *sub-par performance* in speech recognition [33] and autonomous driving [34], opting for the current design hampers further performance improvements.

**End-to-end training of knowledge extraction.** This work takes a fundamentally different approach where we leverage *end-to-end neural network* to perform knowledge extraction for federated learning. First, by using *neural networks* over the conventional approach, the knowledge extraction framework can benefit from the generalization power of the neural networks. This not only makes the knowledge extraction perform better clustering of the clients but also be more robust to the dynamic settings of real-world federated learning. Furthermore, by taking an *end-to-end* design, the knowledge extraction framework can learn richer features that lead to better clustering of the clients.

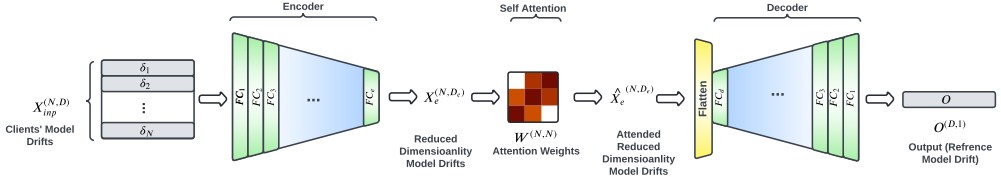

Figure 2: Attention Based Clustering (ABC) network architecture.

**Adopting attention based on analogy to language understanding.** The attention mechanism is a learnable layer which is designed to simulate attention operation (assigning mutual weight for pair of features). It performs an attention operation to its input called *Value* matrix by dot producting it to a matrix of weights generated from its other two inputs *Key* and *Query*. In this work, we use *scaled dot-product self attention* [15], in which weights are calculate as shown in Equation 1.

$$W(Q, K) = \text{softmax}(\frac{QK^T}{\sqrt{D_k}}) \tag{1}$$

This work sets out to adopt an *attention mechanism* in knowledge extraction for federated learning. Inspired by its prevalent adoption in language understanding in which the attention learns the semantic correlation of words in a sentence [3], we utilize the attention mechanism to learn the similarity of private dataset distributions in the context of a reference dataset. The innovation is that the attention mechanism captures the similarities and patterns of a batch of clients' model drifts (received by the central server in each communication round) in a low-dimensional latent space, similar to the way it captures the mutual relation of a batch of words (a sentence). Extracting feature dependencies even in a latent low dimensional space (as is the case in this work) helps in clustering clients.

### 4.2  ABC: Attention Based Clustering

**Architecture.**  The proposed architecture for performing **A**ttention-**B**ased **C**lustering (ABC) consists of three main components as can be seen in Figure 2. The first part is the encoder which is a stack of linear layers each followed by ReLU for reducing the dimension of inputs. The input to the network is a fat two-dimensional

matrix, and the encoder makes it thinner by performing dimensionality reduction The second component is a self attention mechanism which attends to this thinned two-dimensional matrix. The third part is a stack of layers for flattening the output of self attention (attended version of the thinned input) to combine the information in two dimensions into one dimension, and a conventional decoder (like normal autoencoders). The final output of the network is a one-dimensional vector with the same dimensionality as the input matrix width.

**Training.** Figure 3a summarizes the overall formulation of `ABC` training. The `ABC` network is placed in the central server observing the distributed training and client updates in each communication round. Between any two communication round, the model in each client experiences a change (client model drift), and the model in the central server is trained with the same number of training steps as in clients, on the reference dataset The change of model parameters as a result of training on the reference dataset is called reference model drift We formulate the learning problem as a regression problem, in which the network tries to reconstruct the reference model drift, from a batch of clients' model drifts (formed as a two-dimensional matrix) received in each communication round. We use the mean-squared-error loss function for training the network end-to-end. Although we formulate the learning problem as a regression problem for reconstructing reference model drifts from clients' model drifts, we use this network for clustering with the learned weight matrix of the attention mechanism. Overall, this formulation results in *an encoder for effective dimensionality reduction* and *a clustering mechanism from the weight matrix of the attention.*

**Deployment.** Figure 3b summarizes the overall formulation of `ABC` deployment. Once the training phase is finished, `ABC` network is ready to be deployed, without the need to be re-trained or fine-tuned in case of distribution shifts. When `ABC` netwrok receives a set of model drifts, the attention mechanism generates a weight matrix $W^{(N,N)}$, with $N$ the number of clients. Conceptually, element in row $i$ and column $j$ of $W^{(N,N)}$ represent a number, modeling the dependency of client $i$ and $j$. We monitor the sum of rows of this matrix which we call score vector $S^{(1,N)}$, thus element $i$ of $S^{(1,N)}$ is the sum of dependency of client $i$ with all other clients and is called $i_{th}$ client score. These client scores generated by `ABC` network, cluster clients in only one dimension, which is a severe degree of dimensionality reduction while preserving clustering patterns. It is worth mentioning that, it is not necessary to use `ABC` network in each communication round to check clients' clusters (check-in stage), as the frequency of using it depends on how often clients' datasets change. Finally, this client scores $S^{(1,N)}$ are used to pick participants in each communication round. Assuming the resources allow $K$ participants per communication round, we compute the average of all clients' scores ($S^{(1,N)}$) called *averaged center of clusters*, and pick participants such that (1) the average score of participants in each communication round is roughly equal to *averaged center of clusters*, (2) diversify participants as much as possible. Under a specific constraint for the number of participants, this policy ensures that the model under training does not get biased toward any distribution (which is a major reason for wiggliness of the learning curve in federated learning). The `ABC` network is only used for inference in check-in stages, so it does not introduce any major overhead in the regular process of federated learning (less than $1\%$ of training time with `ABC` deployed)

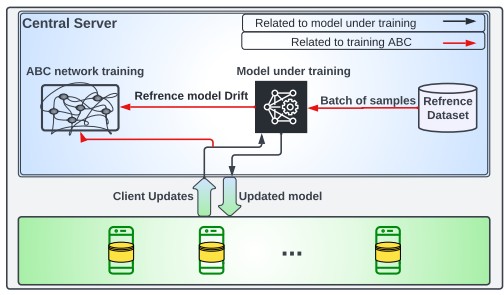

(a) Training phase of `ABC`.

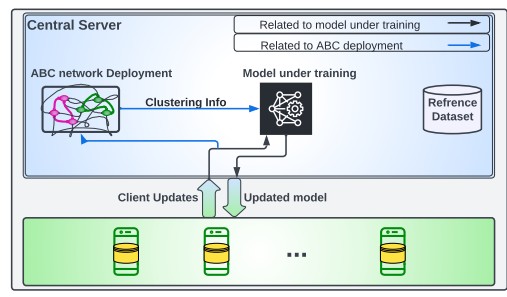

(b) Deployment phase of `ABC`.

Figure 3: The training phase of `ABC`. Network weights are frozen once the training phase is finished, then `ABC` is ready to be deployed, without the need to be re-trained or fine-tuned in case of distribution shifts.

**Implementation details.** We refer to a federated learning session as a full cycle of training a model in federated learning (by any method) until achieving a target accuracy. In our implementation, each federated learning session's configuration consists of the type of dataset to be used in clients (EMNIST, CIFAR10, or FMNIST), the clients' settings (distribution of datasets in clients), the number of participants in each communication round, number of clients, frequency of performing a communication round, learning hyperparameters (optimizer parameters, batch size, and model parameter initialization, random seed). Before starting any federated learning session, all of these configurations must be specified. For generating a specific distribution from a dataset (say EMNIST) to be used in a client, we first determine the number of samples

needed from each class, and then perform uniform sampling from each class as many times as needed to achieve the distribution.

Details about the hyperparameters and the exact setup are included in Appendix C.

## 5 Experiments

In this section, we show the effectiveness of ABC for reducing communication rounds under clients' distribution shift compared to the baseline FedAvg. **Setup.** We evaluate the performance improvements of ABC in terms of communication reduction with FedAvg [9] as the baseline, which is a widely used federated learning method. Each federated learning session is designed as an stand-alone Jupyter notebook that can be used by only setting configuration parameters.

**Client settings.** We define a couple of clients' dataset distribution settings, that we use in our federated learning session. As a reminder, we refer to clients' setting in a federated learning session as the setting of distributions of clients' datasets. Our general approach is training the ABC network under an specific client setting, and evaluating it in that same setting, and other settings for showing the robustness. Each of the pie charts in Figure 4a, 4b, and 4c, shows a specific client setting.

For instance, in Figure 4c which shows 5C setting, we can see that clients are divided into five different clusters, and a client in each cluster has a dataset consists of a specific dominant label (showed in the figure

**Training ABC network.** We train the ABC architecture with the procedure mentioned in section 3.4 for three datasets (EMNIST, FMNIST, CIFAR10) each under 2C-1 setting. Each client consists of $500$ samples ($1000$ in case of CIFAR10)

**Algorithm 1** ABC Deployment

**Initialize**: max epoch, num batches, local models, local datasets, central model, K, T
**Output**: central model

1: epoch $\leftarrow$ 1
2: check in flag $\leftarrow$ True {*for controlling the frequency of using ABC network*}
3:
4: **while** epoch $\leq$ max epoch **do**
5:    **for** $b = 1$ to num batches **do**
6:       local models = Train(local models) {*trains each client's model for one batch of data*}
7:       **if** b $\% \frac{\text{num bathces}}{\text{T}} = 0$ **then**
8:          **if** check in flag $=$ True **then**
9:             client scores $\leftarrow$ ABC(local models)
10:          **end if**
11:          active clients $\leftarrow$ Pick(local models, client scores, K) {*picking up participants based on the policy*}
12:          central model $\leftarrow$ Average(active clients)
13:       **end if**
14:    **end for**
15: **end while**

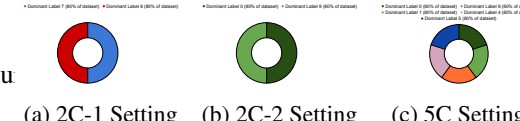

(a) 2C-1 Setting    (b) 2C-2 Setting    (c) 5C Setting

Figure 4: Clients' settings for the experiments.

**Deploying ABC network:** After training the ABC architecture under 2C-1 setting, we use Algorithm 1 to do federated learning sessions under 2C-1, 2C-2, and 5C settings. We use the 2C-2 setting to show the robustness of trained ABC architecture in dealing with a totally new setting in terms of client distributions, but with the same number of clusters. The 5C setting is designed for showing the robustness of the trained ABC architecture when there are more clusters of clients. We push the robustness evaluation even harder by evaluating the trained ABC network in a setting in which all the clients datasets are entirely changed compared to the training time. we deploy ABC and run federated learning sessions under 2C-1, 2C-2, and 5C setting for each dataset ($3 \times 3 = 9$ total number of deployment federated learning sessions). For each of these deployment sessions (9 sessions) we visualize client scores ($S^{(1,N)}$) to verify clusters of similar clients. Finally, we run multiple full blown federated learning sessions for three datasets (EMNIST, FMNIST, CIFAR10), under three settings (to simulate distribution shift, each once with FedAvg and once with ABC in deployment phase. Each of the mentioned federated learning sessions are executed once with $N = 10$ clients with $K = 2$ participant per communication round, and also with $N = 50$ with $K = 2$ and $K = 10$. This adds up to a total of $54$ execution of federated learning sessions. As a extreme case of robustness evaluation, we deploy the ABC network trained on EMNIST, to all the mentioned federated learning sessions but with FMNIST dataset in all clients, which added 18 more execution of federated learning sessions ($18 + 54 = 72$)

**Cluster visualization.** As mentioned before, ABC generates a list of client scores $S^{(1,N)}$ when ever a batch of received model drifts are passed through it in the central server. The trained ABC network under 2C-1 setting is deployed and evaluated under all other three settings with three datasets under study (9 experiments). client scores for some of these experiments can be found in Figure 5. Further cluster visualization plots (under different deployment settings) are available in Appendix A. In Figure 5, horizontal axis shows the id of each client, and the vertical axis shows the score assigned to each client by ABC network under deployment. As is observed in Figure 5, ABC successfully discriminate clusters of similar clients in terms of their distribution in the lowest dimension possible (one dimension - assigning a single number to each client) These two properties of this clustering method which are (1) severe dimensionality reduction while preserving clustering information (2) robustness to distribution shifts, makes it a powerful clustering method.

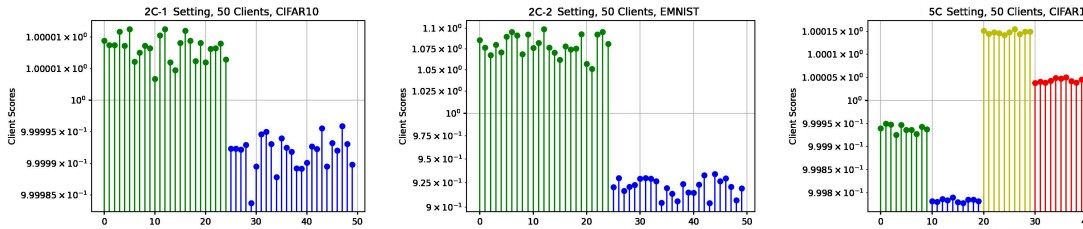

Figure 5: Plots show client scores generated by the `ABC` network under three different deployment settings (2C-1, 2C-2, and 5C) for three datasets.

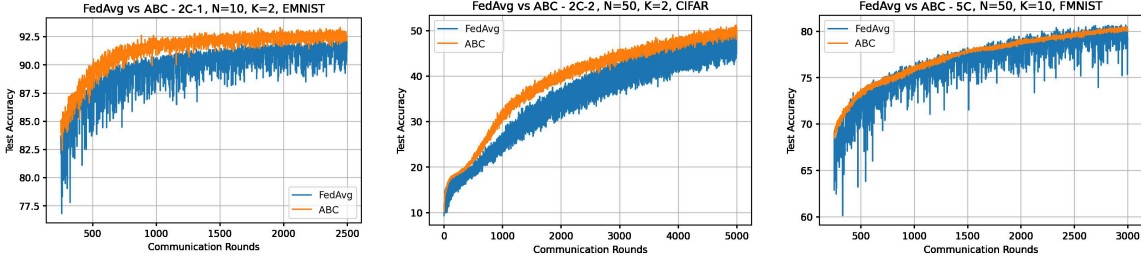

Figure 6: Plots show test accuracy curves of FedAvg and `ABC` in three different federated learning sessions without re-training or fine-tuning the `ABC` network.

**Communication cost analysis.** Having this knowledge (clusters) about clients, we can use it to devise the optimum policy for accelerating federated learning In Table 1, number of communication rounds for many federated learning sessions (including all the clients' settings, datasets, different number of clients, different number of participants per communication round and their different combinations) can be seen. The test accuracy of the model under training is the criteria for terminating the session (45% for CIFAR10, 75% for EMNIST, and 70% for FMNIST).

Also, the model under training in all federated leaning sessions is a CNN with a fully convolutional backbone with ReLU non-linearity, and the head of the network is a stack of linear layers. Results in Table 1 shows that deploying `ABC` network, effectively decrease the communication cost (up to 75% for MNIST and FMNIST, and up to 45% for CIFAR10). Evidently, in presence of distribution shifts, the policy devised using the clustering knowledge from `ABC` is effective in reducing communication rounds compared to the baseline, so it is robust to distribution shifts. As a more extreme case of policy robustness evaluation, we deploy `ABC` network trained on MNIST to various settings in which clients' datasets are FMNIST (It is indicated by FMNIST* in table 1), and `ABC` is still superior to FedAvg.

| Comparison of Communication Rounds | | N=10 | | N=50 | | | |
| --- | --- | --- | --- | --- | --- | --- | --- |
| | | K=2 | | K=2 | | K=10 | |
| | | FedAvg | ABC | FedAvg | ABC | FedAvg | ABC |
| EMNIST | 2C-1 | Not Converged | 967 | 2980 | 738 | 1725 | 1363 |
| | 2C-2 | 3413 | 804 | 2036 | 1015 | 1191 | 1086 |
| | 5C | 834 | 522 | 1516 | 1297 | 865 | 731 |
| FMNIST | 2C-1 | 1606 | 468 | 2718 | 887 | 1134 | 600 |
| | 2C-2 | 913 | 344 | 1151 | 776 | 932 | 580 |
| | 5C | 2474 | 804 | Not Converged | 1697 | 703 | 501 |
| CIFAR10 | 2C-1 | Not Converged | 3908 | Not Converged | 3419 | 4708 | 3058 |
| | 2C-2 | Not Converged | 4986 | Not Converged | 3666 | 4195 | 3712 |
| | 5C | Not Converged | 4997 | Not Converged | 4993 | 3225 | 2204 |
| FMNIST* | 2C-1 | 1606 | 523 | 1516 | 689 | 1134 | 600 |
| | 2C-2 | 913 | 397 | 2718 | 820 | 932 | 580 |
| | 5C | 2474 | 2109 | 1151 | 1406 | 703 | 501 |

Table 1: Number of communication rounds until a target test accuracy is achieved. For each dataset, the `ABC` network is trained under 2C-1 setting. For FMNIST*, `ABC` network is trained under 2C-1 setting of EMNIST dataset.

To better understand how devising this policy based on `ABC` clustering information decreases communication cost, we plot test accuracy of the model under train in three federated learning sessions 6 (more plots available in Appendix B). The learning dynamics in Figure 6 shows that the accuracy curve of `ABC` is much more stable and less wiggly than its counterpart FedAvg. The fluctuations in learning curve of FedAvg is a direct result of the occasional undesired bias of model under training, towards specific clients throughout the training. To minimize this effect, `ABC` adopt a policy in which all the clusters are equally involved (By keeping the average client score of participants in each communication round stable) throughout the training progress This more stable and less wiggly learning curve results in faster convergence of the model under training, thus reduces communication rounds and accelerates federated learning.

## 6   Conclusion

We present attention-based knowledge extraction to improve the policy for choosing participants of each communication round in federated learning. The paper uniquely explores the use of *end-to-end* knowledge extraction with *attention* to better cluster the clients. Extensive experiments on different datasets with variations of client settings show that attention-based knowledge extraction significantly reduces the communication rounds in federated learning, and it is robust to dynamic settings,

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
