# OpenReview forum: "Accelerating Federated Learning Through Attention on Local Model Updates"
_NeurIPS.cc/2022/Workshop/Federated_Learning — FL-NeurIPS 2022 Poster_

### Official Review · Reviewer_q9MX · 2022-10-07

The paper proposes a strategy for sampling clients in federated learning that clusters clients using a novel attention-based clustering and picks clients to cover as many clusters as possible. This approach achieves a similar test accuracy as standard federated averaging with substantially fewer communication rounds, especially when local datasets are heterogeneous.

Using an attention mechanism on the drifts encoded through an autoencoder is definitely a neat idea and the empirical results show gains over vanilla federated averaging on three datasets. There are two main questions the paper leaves unanswered though: (i) how much better is the proposed clustering technique over existing ones (e.g., references [22, 20, 16] from the paper), and (ii) how does picking clients smartly (as proposed in this paper) compare to picking communication rounds smartly (as in [1])?
Moreover, a comparison to approaches more suitable to heterogeneous data (e.g., [2, 3]) would be interesting.

Despite these shortcomings, I think this is a decent paper and I tend towards acceptance.

Detailed comments:
- the text around Fig. 4 is broken
- it would have been great if a link to the jupyter notebooks mentioned in the experiment could have been provided, e.g., via an anonymized github repository (https://anonymous.4open.science/)


[1] Kamp, Michael, et al. "Efficient decentralized deep learning by dynamic model averaging." Joint European conference on machine learning and knowledge discovery in databases. Springer, Cham, 2018.
[2] Li, Tian, et al. "Federated optimization in heterogeneous networks." Proceedings of Machine Learning and Systems 2 (2020): 429-450.
[3] Karimireddy, Sai Praneeth, et al. "Scaffold: Stochastic controlled averaging for federated learning." International Conference on Machine Learning. PMLR, 2020.

---

### Official Review · Reviewer_fyGS · 2022-10-18
**FL with clustering.**

This paper considers the federated learning problem with non-iid data. Because of communication limitations, a sub-set of clients participate in learning and communication at each round of FL. For the client selection process, this paper leverage client clusters that are extracted from an attention-based mechanism. Experiments show that the proposed algorithm outperforms conventional FL methods.

Pros.

Attention-based clustering is a promising direction to generate client clusters.

Cons.

The performance improvement is not significant, although the setting assumes an artificial cluster structure.

There could be more possible methods that can utilize client clients to enhance FedAvg. More discussions might be necessary.

---

### Official Review · Reviewer_ztXt · 2022-10-19
**concern about using attention in parameter space**

The author proposed an attention based method  seeking to solve data distributional shift problem in federated learning which will reduce training training, performance and often lead to sub-optimal solution due to local minimum.

To address this problem, the authors utilize an attention mechanism to understand patterns and clustering of parametric changes between different updating rounds, with a reference model implemented. The author conducted various empirical experiments to show the proposed method has superior performance.

The key concern I have about this work is the attention and all operation happen in parameter space. It is important to point out that in relatively complex model , parameter space are not equal to function space ( i.e two model with different parameters may lead to the same mapping function). I am not sure if the drift in the parameter space is a good thing to use. The improvement in performance may not generalize to deep learning models with various architecture. Instead, in representation space of data points or their aggregation might be better

In addition, the fact the authors uses a reference central data set may not be realistic in real-world FL setting

---

### Decision · Program_Chairs · 2022-10-20

Accept (Poster)